# Stroke outcome assessment: Optimizing cutoff scores for the Longshi Scale, modified Rankin Scale and Barthel Index

**Mingchao Zhou**[1☯], **Xiangxiang Liu**[1☯], **Fubing Zha**[1], **Fang Liu**[1], **Jing Zhou**[1], **Meiling Huang**[1], **Wei Luo**[1], **Weihao Li**[1], **Yuan Chen**[1,2], **Sheng Qu**[1,3], **Kaiwen Xue**[1,3], **Wanqi Fu**[1], **Yulong Wang**[1] *

**1** The First Affiliated Hospital of Shenzhen University, Shenzhen Second People's Hospital, Shenzhen, Guangdong, China, **2** Medical College, Shantou University, Shantou, Guangdong, China, **3** Shandong University of Traditional Chinese Medicine, Jinan, Shandong, China

☯ These authors contributed equally to this work.
* ylwang668@163.com

**Data Availability Statement:** All relevant data are within the manuscript and its Supporting Information files.

## Abstract

The Longshi Scale, a visual-based scale, is reliable and valid in activity assessment, but lacks cutoff definition corresponding to classical scales such as the modified Rankin Scale and Barthel Index. Therefore, this study aimed to investigate the relationships of the Longshi Scale with the modified Rankin Scale and Barthel Index and optimize cutoff scores of these scales in stroke outcomes assessment. This is a cross-sectional study. Stroke patients were measured concurrently by the Longshi scale, modified Rankin Scale and Barthel Index. Kruskal-Wallis test and Spearman correlation analysis were used to analyze the differences and associations among the three scales. The receiver operating characteristic curve was performed to determine the optimal cutoff scores. A total of 5475 stroke patients (67.3% ischemic) were included in this study. There are close relationships of the Longshi Scale with adjusted modified Rankin Scale and Barthel Index ($r$ = -0.861, 0.922; $p$<0.001, <0.001; respectively). The activity levels assessed by adjusted modified Rankin Scale and Barthel Index among different Longshi scale grades were significantly different ($\chi^2$:4217.27, 4676.55; $p$<0.001, <0.001; respectively). The optimal cutoff scores were adjusted modified Rankin Scale 4, 3, 3, 3, 2 for Longshi scale grade 2 to 6 (sensitivity%: 96.12, 70.24, 89.10, 96.80, 86.23, specificity%: 72.72, 98.29, 93.81, 79.82, 92.89, respectively), and Barthel Index 15, 45, 60, 75, 80 for Longshi scale grade 2 to 6 (sensitivity%: 92.54, 89.28, 91.32, 90.30, 95.65, specificity%: 95.48, 89.51, 94.02, 90.41, 90.62, respectively). In conclusion, the classification of Longshi Scale is consistent with those of modified Rankin Scale and Barthel Index. We recommend the Longshi Scale as an effective supplement for modified Rankin Scale and Barthel Index in assessing the outcome in acute stroke patients.

**Funding:** This work was supported by the Guangdong Basic and Applied Basic Research Foundation (Grant No. 2020A1515111062, http://gdstc.gd.gov.cn) and Shenzhen Second People's Hospital Clinical Research Foundation (Grant No. 20203357021, 20203357019, http://www.szrch.com) to MCZ. The funders had no role in study design, data collection and analysis, decision to publish, or preparation of the manuscript.

**Competing interests:** The authors have declared that no competing interests exist.

## Introduction

Stroke is a leading cause of death worldwide [1]. The rates for all age disability-adjusted life-years after stroke have reached 81% in low- and middle-income countries [2] and have increased by 46.8% in China [3] during recent decades. The total global burden of stroke is increasing and the need of post-stroke rehabilitation is growing [1, 2]. The primary endpoint of stroke rehabilitation was the degree of dependence in the patient's own environment [4]. However, the clinical outcome evaluation of rehabilitation for acute stroke is hindered by lack of consensus among different scales [5].

The most commonly used scales for outcome assessment in stroke patients are the modified Rankin Scale (mRS) and the Barthel Index [6, 7]. The mRS scale was originally designed as a handicap scale but is now more commonly used to assess global disability [8]. There are 7 grades in mRS, ranging from 0 (no symptoms at all) to 6 points (mortality) [9]. The mRS has been proven to be reliable and valid for defining outcomes in patients with stroke [10]. The BI is also a widely used scale for measuring the activities of daily living (ADL). There are 10 items in BI, ranging from 0 (complete dependence of ADL) to 100 points (complete independence of ADL) [11]. The BI has also been proven reliable and valid for evaluating the effectiveness of rehabilitation in stroke patients [12].

However, the mRS and BI scales mainly focus on assessing the body function and activity but limited in assessing the family and social participation ability. The World Health Organization International Classification of Functioning categorizes the consequences of disease into 3 different dimensions: body function impairments, activity limitations, and participation restrictions [13]. Therefore, it needs to develop new assessment method to assess the participation restrictions.

The Longshi scale was originally designed as a visual-based scale to assess activity ability of functionally disabled patients [14]. Disabilities were divided into three groups (i. e. bedridden, domestic, and community groups) in LS scale, which not only reflecting the basic ADL but also the family and social participation functions. Our preliminary studies indicated that LS was valid in assessing the ADL of functionally disabled patients, and LS had good inter-rater and intra-rater reliability [14, 15]. These findings suggest that LS may have wider applications than mRS and BI. However, the relationships of LS with mRS and BI are still unclear. This study aimed to investigate the relationship of LS with mRS and BI, and determine the cut-off scores of the mRS and BI for all LS grades for consistently defining the outcome in acute stroke patients.

## Materials and methods

### Study design

This is a multicenter cross-sectional study. The Data was gathered from a clinical study "the Promotion of Application research of Longshi Scale", managed by the Rehabilitation department of Shenzhen Second People's Hospital, shenzhen, China, and performed at the Rehabilitation Department of 88 hospitals and community rehabilitation centers around China from December 2018 to May 2020. The study protocol was approved by the Ethics Committee of Shenzhen Second People's Hospital (No. 20180926006) and was registered in the Chinese Clinical Trial Registry (ChiCTR-2000034067).

### Study population

The study population was selected from the multicenter clinical study according to the inclusion and exclusion criteria. The inclusion criteria were as follows: cerebral stroke, including

hemorrhages and infarcts diagnosed by computed tomography or magnetic resonance imaging; age over 18 years old; duration of stroke less than 3 month. The exclusion criteria were as follows: (1) patients with brain trauma, brain tumor, or brain abscess; (2) patients with spinal cord injury, motor neuron disease, or Parkinson's disease; (3) patients with autism, Alzheimer, or mental retardation; (4) patients with fracture, lumbar disc herniation, knee cruciate ligament tear or burn injury; (5) patients with aphasia, unconsciousness, Severe anxiety and depression; (6) patients who do not cooperate with the evaluation. Accordingly, a total of 7286 hospitalized stroke patients were screened for eligibility. After excluding cases that do not meet the inclusion criteria, including patients with duration of stroke over 3 month (n = 1763) and important data missing (n = 48), 5475 patients were included in this study. Each patient was provided details about the study before the measurement, after which patients provided written informed consent.

## Baseline characteristics

Baseline characteristics were collected through a questionnaire during hospitalization, which included (1) demographic information: name, gender, and age; (2) disease status: type of stroke, duration of disease, and frequency of attacks; (3) history of comorbidities: hypertension, diabetes; heart disease; (4) and living habits: smoking and alcohol drinking. All characteristics were collected face to face by professional therapists. In order to minimise bias, two professional therapists of each hospital or community rehabilitation center were gathered for uniform training before data collection. And during the study period, the data was supervised by two quality controllers to ensure the authenticity and security. All co-authors had access to information that could identify individual participants during or after data collection. But any modification of the recorded data was prohibited.

## ADL measurement

ADL levels of the subjects were measured in turn by mRS, BI, and LS scales on the same day. Data were entered through an application Kangfu Kuaixian (Co. Ltd. Yilanda. Shenzhen, China) and saved on the website (http://www.yilanda.cn) immediately after the measurements were finished. The method of the mRS and BI measurement refers to the previous literature [6].

According to the findings of Uyttenboogaart et al, many mRS grades were not correctly classified [5]. About 11% - 14% of patients with mRS 1 and 28% - 43% of patients with mRS 2 were dependent for at least one activity of the BI items, and 27% - 31% of patients with mRS 3 were dependent for walking. Therefore, they suggested that patients with mRS scores of 1 or 2, that were dependent for any of the BI items (except for bladder and bowel control), should be assigned with mRS score 3; patients with mRS scores 1, 2, or 3 who walked dependently, should be assigned with mRS score 4; and patients with mRS score 4 who walked independently should be assigned with mRS score 3 [5]. Accordingly, the recorded mRS data in our study would be adjusted if needed before analysis.

LS was originally a visual-based scale designed by the Rehabilitation Department of Shenzhen Second People's Hospital to assess ADL of functionally disabled patients. The measurement protocol of LS is outlined in detail in a previous study [14]. According to the LS scale classification system (Fig 1), disabled patients are first divided into bedridden and non-bedridden groups by their ability to independently get off the bed. Second, the non-bedridden group is further divided into domestic and community groups by their ability to go outdoors independently. Third, according to the LS scale assessment system (Fig 2), subjects in each group (bedridden, domestic, and community groups) were further evaluated by three items, showed

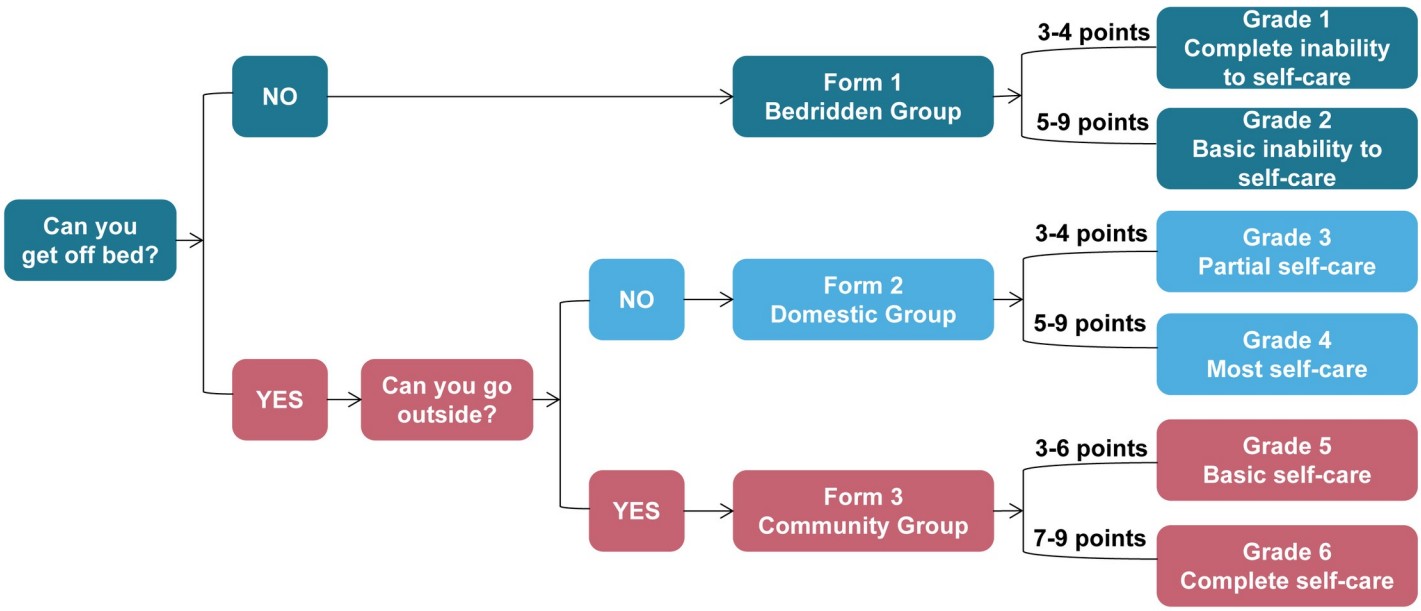

**Fig 1. The Longshi Scale classification system.**

by pictures. Each item consist of there levels, counting 1, 2 and 3 points, 1 for maximum to fully dependent, 2 for partially independent, and 3 for maximum to fully independent. Therefore, the activity levels of each group range from 3 to 9 points [14]. The details of the definitions of each group are as follows:

1. bedridden group: Subjects who have the ability to get off the bed independently. Activity levels are assessed for feeding, bladder and bowel management, and entertainment;

2. domestic group: Subjects who have the ability to get off the bed independently but unable to go outdoors independently. Activity levels were assessed for toileting, grooming and bathing, and housework;

3. community group: Subjects who have the ability to go outdoors independently. Activity levels were assessed for community mobility, shopping, and social participation.

Finally, the three groups of LS were divided into six grades according to the total scores for all items, The details of the definitions of each grade are as follows (Fig 1):

1. grade 1: Complete inability to self-care. Subjects in the bedridden group with scores 3 to 4;

2. grade 2: Basic inability to self-care. Subjects in the bedridden group with scores 5 to 9;

3. grade 3: Partial self-care. Subjects in the domestic group with scores 3 to 4;

4. grade 4: most self-care.Subjects in the domestic group with scores 5 to 9;

5. grade 5: basic self-care. Subjects in the community group with scores 3 to 6;

6. grade 6: complete self-care. Subjects in the community group with scores 7 to 9;

Our previous study has demonstrated that LS was both reliable and valid for disability assessment [14]. In addition, LS became the national standard of ADL measurement in China in 2019. (No. GB/T 37103–2018).

**Fig 2. The Longshi Scale assessment system.**

## Statistical analysis

All analyses were performed using SPSS 21.0 (IBM, New York, USA). Mean ± standard deviation or median and interquartile were used to describe continuous variables. Individuals with important data missing were excluded from the analysis. Descriptive statistics were performed to examine the distribution of the mRS and LS according to BI scores. Kruskal-Wallis test was used to analysis the differences of mRS and BI among different LS grades. Spearman correlation analysis was performed to investigate the association of LS with the BI and mRS, which could reflect the criterion validity of LS.

The LS grades were dichotomized into two categories distinguishing the "favorable outcomes" and "unfavorable outcomes" at all cutoff points. For example, to calculate mRS and BI cutoff scores in LS grade 6, LS grades 1 to 5 were set as "unfavorable outcomes" and LS grade 6 was set as "favorable outcomes." The sensitivities, specificities of the mRS and BI scores for LS

grades were calculated at these cutoff points. The receiver operating characteristic curve (ROC) was used to investigate the relationship between sensitivity and specificity. The optimal cutoff scores of mRS and BI in LS grades were determined as the score with the highest sum of sensitivity and specificity. the area under the curve (AUC) was calculated to identify the discrimination potential of the mRS and BI cutoff score in the LS grades. A *P*-value < 0.05 was considered statistically significant.

## Results

### Clinical characteristics of stroke patients

The patients' baseline characteristics are presented in Table 1. Of all the included patients, 3686 (67.3%) had an ischemic stroke and 1789 (32.7%) a hemorrhagic stroke; 3485 (63.7%) were male. The mean age was 61.56 ± 20.07 years. The median of duration of stroke was 8 days, indicating the patients were in the acute phase of stroke.

### The distribution of Longshi Scale, mRS and BI

The distribution of LS, mRS, and BI were presented in Table 2. A total of 2985 (54.9%), 1583 (28.9%), and 907 (16.6%) individuals belonged to the bedridden, domestic, and

**Table 1. Clinical characteristics of stroke patients.**

| Characteristics (n = 5475) | Statistics |
|---|---|
| Age (Mean+SD, years) | 61.56 ± 20.07 |
| Gender (n, %) | |
| Male | 3483 (63.6%) |
| Female | 1992 (36.4%) |
| Duration of stroke (M, IQR, days) | 8.00 (2.00, 30.00) |
| Type of stroke (n, %) | |
| Ischemic | 3686(67.3%) |
| Hemorrhagic | 1789 (32.7%) |
| Frequency of attacks (n, %) | |
| 1 | 362 (82.6%) |
| ≥2 | 76 (17.4%) |
| Cardiac disease (n, %) | |
| No | 4388 (80.1%) |
| Yes | 1087 (19.9%) |
| History of Hypertension (n, %) | |
| No | 1581 (28.9%) |
| Yes | 3894 (71.7%) |
| History of Diabetes (n, %) | |
| No | 4096 (74.8%) |
| Yes | 1379 (25.2%) |
| History of smoke (n, %) | |
| No | 4104 (75.0%) |
| Yes | 1371 (25.0%) |
| History of alcohol drinking (n, %) | |
| No | 4879 (89.1%) |
| Yes | 596 (10.9%) |

SD: Standard Deviation; M: median, IQR: Interquartile Range.

**Table 2. The distribution of LS scale, recorded and adjusted mRS and BI.**

| LS groups | mRS grades | Recorded mRS number (n, %) | Adjusted mRS number (n, %) | Adjusted mRS number (n) | BI levels corresponding to Adjusted mRS (mean ±SD, median) |
|---|---|---|---|---|---|
| Bedridden group | mRS 0 | 27 (0.49) | 0 (0.00) | 27 | - |
| | mRS 1 | 14 (0.26) | 0 (0.00) | 14 | - |
| | mRS 2 | 20 (0.37) | 0 (0.00) | 20 | - |
| | mRS 3 | 131 (2.39) | 51 (0.93) | 80 | 61.96±13.16 (60) |
| | mRS 4 | 1790 (32.69) | 1925 (35.16) | 135 | 28.89±16.56 (30) |
| | mRS 5 | 1003 (18.32) | 1009 (18.43) | 6 | 3.40±7.24 (0) |
| Domestic group | mRS 0 | 10 (0.18) | 5 (0.09) | 5 | 100 (100) |
| | mRS 1 | 88 (1.61) | 77 (1.41) | 11 | 87.01±10.86 (90) |
| | mRS 2 | 192 (3.51) | 154 (2.81) | 38 | 85.26±11.17 (85) |
| | mRS 3 | 636 (11.62) | 635 (11.60) | 1 | 73.11±12.26 (75) |
| | mRS 4 | 645 (11.78) | 701 (12.80) | 56 | 51.91±14.08 (50) |
| | mRS 5 | 12 (0.22) | 11 (0.20) | 1 | 43.18±22.61 (45) |
| Community group | mRS 0 | 57 (1.04) | 49 (0.89) | 8 | 100 (100) |
| | mRS 1 | 364 (6.65) | 362 (6.61) | 2 | 96.96±5.91 (100) |
| | mRS 2 | 256 (4.68) | 288 (5.26) | 32 | 93.80±8.53 (95) |
| | mRS 3 | 183 (3.34) | 179 (3.27) | 4 | 84.47±9.84 (85) |
| | mRS 4 | 44 (0.80) | 29 (0.53) | 15 | 52.24±23.09 (55) |
| | mRS 5 | 3 (0.05) | 0 (0.00) | 3 | - |
| total | | 5475 | 5475 | 458 | 44.58±33.68 (43.7) |

LS: Longshi Scale; BI: Barthel Index; mRS: modified Rankin Scale.

community groups, respectively. Data showed that 27(0.49%), 14(0.26%,) and 20(0.37%) of bedridden patients were assigned as mRS 0, 1, and 2, which were independent for walking and there appear to be quite infeasible. After adjustment, a total of 20.2%, 2.1%, and 4.5% of the patients with mRS 0, 1, and 2 were assigned to the mRS score 3, respectively. A total of 17.0%, 4.1%, 9.0%, and 15.6% patients with mRS 0, 1, 2, and 3 were assigned to mRS score 4, respectively.

After adjustment, the mean value of adjusted mRS and BI for all LS grades were showed in Table 3. The Kruskal-Wallis test showed that there were significant differences in adjusted mRS and BI levels among all LS grades ($\chi^2_{totle}$ = 4217.27, 4676.55; $p<0.001$, $<0.001$). These results indicated that the categories method of LS can effectively distinguish the different ADL levels, no matter corresponding to mRS or BI.

## Correlation analysis of LS with adjusted mRS and BI

The correlation analysis was conducted to analyze the relationship of LS with adjusted mRS and BI (Table 3). It was showed that LS grades were positively associated with BI ($r = 0.925$, $p<0.001$) and negatively associated with adjusted mRS ($r = -0.885$, $p<0.001$). Separately, the scores of three groups of LS (bedridden, domestic, and community groups) were all positive associated with BI ($r = 0.866$, 0.785, 0.691; $p<0.001$, $<0.001$, $<0.001$, respectively) and negative associated with adjusted mRS ($r = -0.668$, -0.658, -0.500; $p<0.001$, $<0.001$, $<0.001$, respectively).

## ROC curves analysis of LS, adjusted mRS and BI

The optimal adjusted mRS and BI cutoff scores in relation to LS grades and the sensitivities, specificities of each cutoff are shown in Fig 3. The optimal cutoff score were mRS 4, BI 15 for

**Table 3. The correlations among LS with adjusted mRS and BI.**

| Grades of LS | n | Adjusted mRS | | | BI | | |
|---|---|---|---|---|---|---|---|
| | | mean±SD | r | p-value | mean±SD | r | p-value |
| Bedridden group | | | -0.668 | <0.001 | | 0.866 | <0.001 |
| Grade 1 | 1173 | 4.73±0.45[a] | | | 3.01±6.38[b] | | |
| Grade 2 | 1812 | 4.06±0.33[a] | | | 32.38±15.68[b] | | |
| Domestic group | | | -0.658 | <0.001 | | 0.785 | <0.001 |
| Grade 3 | 728 | 3.73±0.57[a] | | | 52.33±13.71[b] | | |
| Grade 4 | 855 | 2.85±0.85[a] | | | 76.64±14.18[b] | | |
| Community group | | | -0.500 | <0.001 | | 0.691 | <0.001 |
| Grade 5 | 217 | 2.47±0.91[a] | | | 79.52±16.62[b] | | |
| Grade 6 | 690 | 1.53±0.83[a] | | | 96.22±6.46[b] | | |
| Total | | | -0.885 | <0.001 | | 0.922 | <0.001 |

LS: Longshi Scale; BI: Barthel Index; mRS: modified Rankin Scale; SD: Standard Deviation; r: Spearman correlation coefficient, $\chi^2$: chi-square; $\chi^2_{1,2}$: chi-square of mRS or BI between LS grade 1and 2, and the rest can be understand in the same manner; $\chi^2_{totle}$: chi-square of mRS or BI among all the LS grades.

a: Kruskal-Wallis test in adjusted mRS among different LS grades. $\chi^2_{1,2} = 1296.71$, $p_{1,2} < 0.001$; $\chi^2_{2,3} = 292.68$, $p_{2,3} < 0.001$; $\chi^2_{3,4} = 501.23$, $p_{3,4} < 0.001$; $\chi^2_{4,5} = 35.07$, $p_{4,5} < 0.001$; $\chi^2_{5,6} = 156.54$, $p_{5,6} < 0.001$; $\chi^2_{totle} = 4217.27$, $p < 0.001$.

b: Kruskal-Wallis test in BI among different LS grades. $\chi^2_{1,2} = 1296.71$, $p_{1,2} < 0.001$; $\chi^2_{2,3} = 692.26$, $p_{2,3} < 0.001$; $\chi^2_{3,4} = 730.39$, $p_{3,4} < 0.001$; $\chi^2_{4,5} = 15.40$, $p_{4,5} < 0.001$; $\chi^2_{5,6} = 324.81$, $p_{5,6} < 0.001$; $\chi^2_{totle} = 4676.55$, $p < 0.001$.

LS grade 2 (sensitivity:96.12%, 92.54%; specificity:72.72%, 95.48%; AUC: 0.901, 0.983; respectively), mRS 3, BI 45 for LS grade 3 (sensitivity:70.24%, 89.28%; specificity:98.29%, 89.51%; AUC: 0.894, 0.963; respectively), mRS 3, BI 60 for LS grade 4 (sensitivity:89.10%, 91.32%; specificity:93.81%, 94.02%; AUC: 0.942, 0.979; respectively), mRS 3, BI 75 for LS grade 5 (sensitivity:96.80%, 90.30%; specificity:79.82%, 90.41%; AUC: 0.944; 0.960; respectively), mRS 2, BI 80 for LS grade 6 (sensitivity:86.23%, 95.65%; specificity:92.89%, 90.62%; AUC: 0.956; 0.976; respectively).

## Discussion

In this study, we analyzed the relationship of LS with the mRS and BI, and determined the cutoff scores of the mRS and BI for all LS grades in acute stroke patients. We found that the adjusted mRS and BI scales were significant associated with the LS scales. The LS grades were both consistent with the adjusted mRS and BI scores. These findings may help clarify the application value of the LS scales in outcome evaluation in acute stroke patients.

The mRS and BI scales were commonly used clinical scales to measure the outcome of stroke patients. The mRS could reflect the global disability, while the BI mainly assess ADL levels [16]. As the results showed, both the adjusted mRS and BI were incrementally distributed among the three groups as well as the six grades in LS, and the mean levels of adjusted mRS and BI were significantly different among all LS grades, indicating that the LS was effective in identifying different disability and activity levels in stroke patients. Along with the preliminary findings that LS has high repeatability and is valid when used to assess the ADL of functionally disabled patients [14], we demonstrated that the LS is feasible with good reactivity to evaluate clinical outcomes in stroke patients.

Although mRS and BI are widely used to measure the outcome of stroke patients, there is yet no "gold standard" for defining the favorable outcome because of inconsistent findings in different studies [17]. Hacke et al. have reported that BI ≧75 and mRS ≦2 could be used to define a favorable outcome in acute stroke patients (sensitivity: 75.0%, specificity: 97.8%) [18].

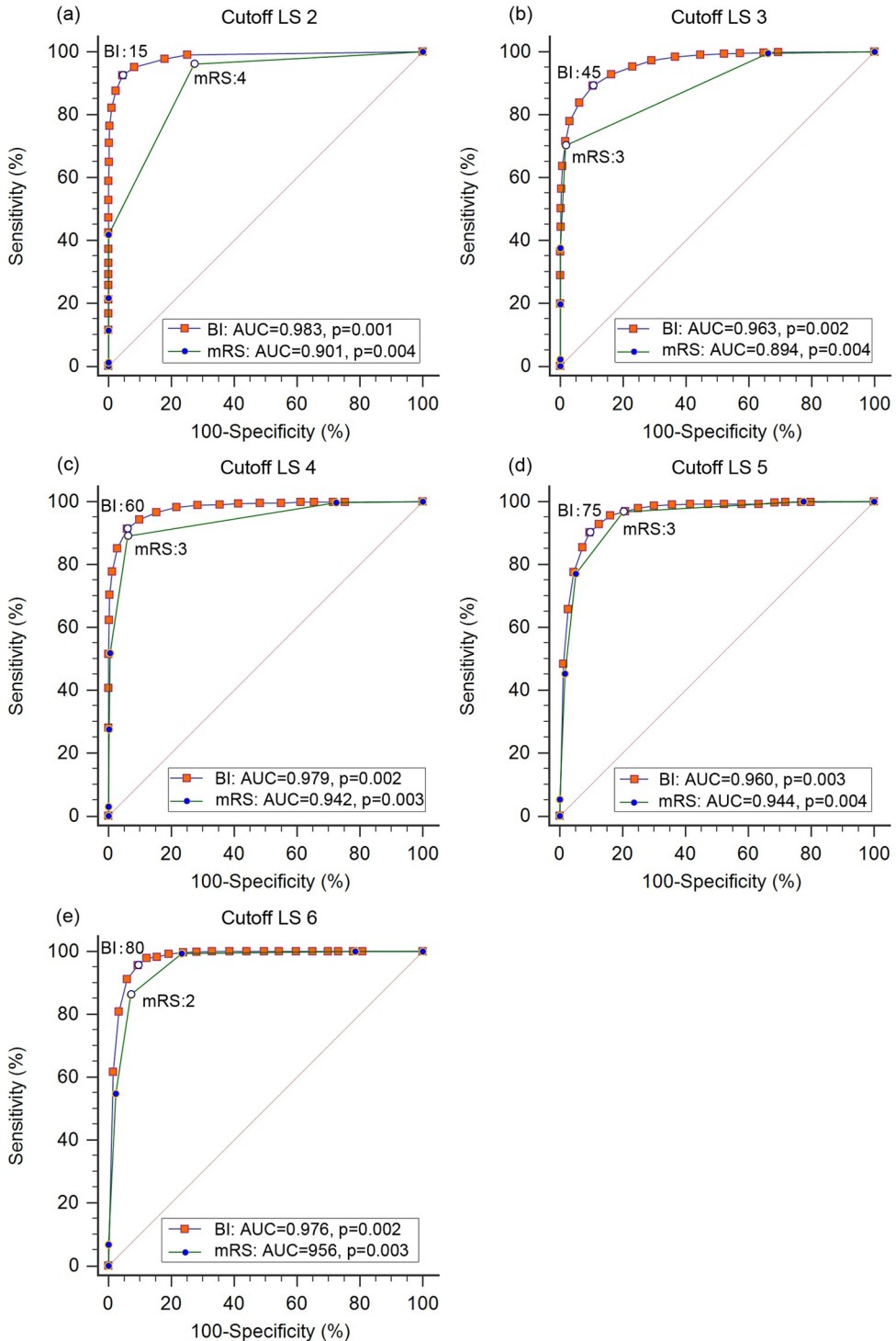

**Fig 3. ROC curves for adjusted mRS and BI cutoff scores in LS grades 2, 3, 4, 5 and 6.** LS: Longshi Scale; mRS: modified Rankin scale; BI: Barthel Index; ROC: Receiver Operating Characteristic; AUC: Area Under Curve. (a) ROC curves for adjusted mRS and BI cutoff scores in LS 2. sensitivity:96.12%, 92.54%; specificity:72.72%, 95.48%, 95.48%; AUC: 0.901, 0.983; respectively; (b) ROC curves for mRS and BI cutoff scores in LS 3. sensitivity:70.24%, 89.28%; specificity:98.29%, 89.51%; AUC: 0.894, 0.963; respectively; (c) ROC curves for mRS and BI cutoff scores in LS 4. sensitivity:89.10%, 91.32%; specificity:93.81%, 94.02%; AUC: 0.942, 0.979; respectively; (d) ROC curves for mRS and BI cutoff scores in LS 5. sensitivity:96.80%, 90.30%; specificity:79.82%, 90.41%; AUC: 0.944; 0.960; respectively; (e) ROC curves for mRS and BI cutoff scores in LS 6. sensitivity:86.23%, 95.65%; specificity:92.89%, 90.62%; AUC: 0.956; 0.976; respectively.

However, Uyttenboogaart et al. found these cutoff scores to be sub-optimal because of the low sensitivity. Further, he suggested that a BI score ≧90 with mRS ≦2 to be the optimal cutoff score to define a favorable outcome (sensitivity: 90.7%, specificity: 88.1%) [16]. Recently, a study of 5,759 stroke patients found the optimal cutoff points of the Korean versions of the MBI (K-MBI) to be 98 (sensitivity: 90.4%; specificity 83.8%) and 94 (sensitivity: 88.5%; specificity: 93.7%) for mRS grades 1 and 2, respectively [19]. Our preliminary study found the optimal cutoff scores of BI to be 100 (sensitivity: 100%; specificity 95.3%) and 100 (sensitivity: 98.1%; specificity 100%) corresponding to the adjusted mRS 1 and 2, respectively [20]. All the above studies were inconsistent in defining the favorable stroke outcome. In this study, we have calculated all the cutoff scores of adjusted mRS and BI for all LS grades and found that LS grade 6 corresponded to mRS 2 as well as BI 80 with high sensitivities and specificities, which may be consistently used to define the favorable outcome in stroke patients in the future.

By comparison, the relationship of LS with mRS and BI were different. LS was more closely correlated with BI than that with mRS (r = 0.922 > │-0.861│). Meanwhile, the cutoff scores of BI for LS grades were distinguishable by different scores, while the cutoff scores of adjusted mRS were both 3 for LS grade 3, 4 and 5, which limited the sensitivity of using mRS scores to correspond to LS grades. This may be due to the subjective nature of the mRS score and lack of clear criteria, which reduced the reliability of the measurement [21]. As we know, there are only 7-point ordinal scores in mRS, which may be insufficient to precisely evaluate patient disability [22].

The distribution of the subjects showed that not all the individuals with mRS 0 or BI 100 belonged to the community group in LS. We speculate that the assessment deviation was one of the possible reasons. Besides, this phenomenon reflects the fact that stroke patients with high levels of mobility and ADL may not have enough ability to return to their social life. Since the primary goal of post-stroke rehabilitation is to help patients reintegrate with their families and social life [23]. Therefore, not only good physical ability but also good cognitive and behavioral ability are required to reach that goal [24].

However, the mRS and BI scales are well used at assessing the activity outcome but limited in assessing the family and social living ability [25]. Fortunately, the LS scale covers both these functions. The assessment items of the LS including feeding, bladder and bowel management, toileting, grooming, and bathing were also measured by the BI. But the family and social participation items, such as entertainment, housework, community mobility, shopping, and social participation, were only measured by the LS rather than by the mRS and BI. Besides, the mRS and BI could only be used by professional healthcare for accurate assessment. Our previous study has found that LS scale had good inter-rater and intra-rater reliability among healthcare professionals and non-professionals including therapists, interns, and personal care aids (ICC$_{1,1}$ = 0.822–0.882 on Day 1; ICC$_{1,1}$ = 0.842–0.899 on Day 7 for inter-rater reliability) [14]. Furthermore, the measurement of the LS scale spends much less time compared with the BI assessment ($p < 0.01$). Therefore, LS has the potential to be served as an effective supplement of mRS and BI when determining the outcome in stroke patients.

According to the ROC analysis results, subjects who want to perform social activities (corresponding to LS ≧5) may need the ADL levels of mRS ≦2 or BI ≧75. For the relative severe cases, at least mRS ≦3 or BI ≧45 were needed to participant in their family life (corresponding to LS ≧3). Especially, the favorable outcome of acute stroke patients could be consistently defined as LS ≧6, adjusted mRS ≦2 and BI≧80. Therefore, the joint application of the three scales may be more conducive when making a comprehensive judgment on the outcome of stroke patients than a single use of each.

There are three limitations to this study. First, patients with a duration of stroke within 3 months were selected in this study, therefore, the optimal cutoff scores of the LS scale may not

be suitable for the sub-acute or chronic stroke patients when defining the favorable outcome, this requires further investigations. Second, as a cross-sectional observational designed research, the reactivity of LS compared with mRS and BI were not answered in this study. Third, patients with aphasia or who do not cooperate with the evaluation were excluded from the study because they have difficulty in completing the assessment, which could translate into a selection bias by leaving out usual scenarios for the application of the scales. Therefore, in the future, we plan to explore the consistency between the assessment results of caregiver and patient to expand the application scenarios of the scales.

## Conclusions

Stroke outcome assessment is hindered by lack of consensus among different scales. The LS scale, a visual-based scale, assessing not only ADL but also family and social participation functions, was closely correlated with mRS and BI. The LS grade $\geqq6$ corresponding to mRS $\leqq2$ and BI $\geqq80$ could be chosen as the optimal cutoff for defining a favorable outcome in acute stroke patients. Therefore, We suggest that the LS could be served as an effective supplement for the mRS and BI in assessing the functional outcome in acute stroke patients.

## Supporting information

**S1 File. The data supporting this work.**
(XLS)

**S2 File. STROBE checklist.**
(DOCX)

## Acknowledgments

The authors thank all the staff members from our institution who actively cooperated in the current research. We especially give our thanks to Miss Chunli Cai (Co. Ltd. Yilanda. Shenzhen, China) for providing us with data integration services.

## Author Contributions

**Conceptualization:** Mingchao Zhou, Xiangxiang Liu, Fubing Zha, Meiling Huang, Wei Luo, Yulong Wang.

**Data curation:** Mingchao Zhou, Xiangxiang Liu, Wanqi Fu.

**Formal analysis:** Mingchao Zhou, Xiangxiang Liu, Fang Liu, Jing Zhou.

**Funding acquisition:** Mingchao Zhou, Yulong Wang.

**Investigation:** Fang Liu, Jing Zhou, Meiling Huang, Wei Luo, Weihao Li, Yuan Chen, Sheng Qu, Kaiwen Xue.

**Methodology:** Mingchao Zhou, Xiangxiang Liu.

**Project administration:** Jing Zhou, Wanqi Fu.

**Supervision:** Fubing Zha, Jing Zhou, Wanqi Fu.

**Writing – original draft:** Mingchao Zhou.

**Writing – review & editing:** Mingchao Zhou, Yulong Wang.

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
