## [Decision Letter · Decision Letter 0]

20 Apr 2021

Stroke outcome assessment: optimizing cutoff scores for the Longshi Scale, modified Rankin Scale and Barthel Index

PONE-D-21-04938

Dear Dr. Wang,

We’re pleased to inform you that your manuscript has been judged scientifically suitable for publication and will be formally accepted for publication once it meets all outstanding technical requirements.

Kind regards,

Miguel A. Barboza, MD, MSc

Academic Editor

PLOS ONE

Additional Editor Comments (optional):

Reviewers' comments:

Reviewer's Responses to Questions

**Comments to the Author**

1. Is the manuscript technically sound, and do the data support the conclusions?

Reviewer #1: Yes

Reviewer #2: Yes

2. Has the statistical analysis been performed appropriately and rigorously? 

Reviewer #1: Yes

Reviewer #2: I Don't Know

3. Have the authors made all data underlying the findings in their manuscript fully available?

Reviewer #1: Yes

Reviewer #2: Yes

4. Is the manuscript presented in an intelligible fashion and written in standard English?

Reviewer #1: Yes

Reviewer #2: Yes

5. Review Comments to the Author

Reviewer #1: With great pleasure and interest I have reviewed the titled work “Stroke outcome assessment: optimizing cutoff scores for the Longshi Scale, modified Rankin Scale and Barthel Index” of the authors Zhou M et al.

The Longshi Scale, a visual-based scale, is reliable and valid in activity ssessment.

Authors aimed to investigate the relationships of the Longshi Scale with the modified Rankin Scale and Barthel Index and optimize cutoff scores of these scales in stroke outcomes assessment.

With a large sample of patients and with an adequate statistical analysis, the authors specify these cutoffs transversely.

This is an important study. In the real world, many healthcare professionals are unfamiliar with the modified Rankin scale or the Barthel index. Furthermore, many patients cannot be scored due to inconsistencies in the definition of the categories for each of these scales. The Longshi visual scale could provide this information and / or complement the others.

The only observation would be the poor definition of figure 1. No more comments.

Reviewer #2: I appreciate the opportunity to read by first-hand the article submitted by Dr. Zhou and colleagues. I would like to highlight the interest in improving the quality of the neurological and disability evaluation of patients after a stroke. Is known to all, that the wide variety of sequelae and levels of disability that these patients develop makes it complicated to objectify the degree of functionality. Considering the above, the reliability of widely used scales such as the modified Rankin scale (mRS) has been raised (Cerebrovasc Dis. 2010 Jan; 29 (2): 188-93). The effort to develop a larger scale, validate it and compare it to demonstrate its usefulness is valuable. This article seeks to define the relationship between the mRS scale and the Barthel index (BI) with the Longshi Scale. Previously, other authors (J Int Med Res. 2020; 48 (7)) have shown a good correlation of LS with BI, but not with mRS.

As part of the review, I have no further observations to make to the abstract or introduction. In the method segment, patients with aphasia or who do not cooperate with the evaluation are specified as exclusion criteria, this could translate into a selection bias by leaving out usual scenarios for the application of the scales. The large number of bedridden patients classified within low mRS levels stands out, which is why the correction made by the authors is very appropriate, which in turn translates into more appropriate results for the purpose of the publication.

Regarding the results and discussion, I believe they are adequately informed and supported. In the same way, the authors manage to define their objective with a good statistical basis, to establish a cut-off for what would be a good outcome after a stroke.

6. PLOS authors have the option to publish the peer review history of their article (what does this mean?). If published, this will include your full peer review and any attached files.

Reviewer #1: No

Reviewer #2: **Yes: **Alvaro Hernández-Guillén

---

## [Editor Report · Acceptance letter]

4 May 2021

PONE-D-21-04938 

Stroke outcome assessment: optimizing cutoff scores for the Longshi Scale, modified Rankin Scale and Barthel Index 

Dear Dr. Wang:

I'm pleased to inform you that your manuscript has been deemed suitable for publication in PLOS ONE. Congratulations! Your manuscript is now with our production department. 

Kind regards, 

on behalf of

Dr. Miguel A. Barboza 

Academic Editor

PLOS ONE